# Acute Respiratory Distress Syndrome; A Review of Recent Updates and a Glance into the Future

**DOI:** 10.3390/diagnostics13091528

**Published:** 2023-04-24

**Authors:** Husayn F. Ramji, Maida Hafiz, Hiba Hammad Altaq, Syed Talal Hussain, Fawad Chaudry

**Affiliations:** 1University of Oklahoma College of Medicine, University of Oklahoma Health Sciences Center, Oklahoma City, OK 73104, USA; husayn-ramji@ouhsc.edu; 2Hudson College of Public Health, University of Oklahoma Health Sciences Center, Oklahoma City, OK 73104, USA; 3Department of Sleep Medicine, University of Oklahoma Health Sciences Center, Oklahoma City, OK 73104, USA; 4Department of Pulmonary, Critical Care & Sleep Medicine, University of Oklahoma Health Sciences Center, Oklahoma City, OK 73104, USAfawad-chaudry@ouhsc.edu (F.C.)

**Keywords:** acute respiratory distress syndrome, pharmacotherapy in ARDS, metabolomics, mesenchymal stem cell transplantation, gut-lung microbiome

## Abstract

Acute respiratory distress syndrome (ARDS) is a rapidly progressive form of respiratory failure that accounts for 10% of admissions to the ICU and is associated with approximately 40% mortality in severe cases. Despite significant mortality and healthcare burden, the mainstay of management remains supportive care. The recent pandemic of SARS-CoV-2 has re-ignited a worldwide interest in exploring the pathophysiology of ARDS, looking for innovative ideas to treat this disease. Recently, many trials have been published utilizing different pharmacotherapy targets; however, the long-term benefits of these agents remain unknown. Metabolomics profiling and stem cell transplantation offer strong enthusiasm and may completely change the outlook of ARDS management in the near future.

## 1. Introduction

During the 1918–1919 influenza pandemic, a few documented causes of death were “purulent bronchitis”, “fulminating pneumonia”, and “lungs full of hemorrhages” [1,2]. Similarly, in the years leading up to that, patients had succumbed to “wet lung” and “shock lung”, and documentation such as this would have occurred until the 1960s [3]. Today, we would report these causes of death as acute respiratory distress syndrome, or “ARDS”. ARDS is a rapidly progressive form of respiratory failure that occurs as a result of excessive inflammation and accounts for 10% of admissions to the intensive care unit (ICU) [4]. ARDS was not described in the published literature until 1967, but by then it had already left its mark on the medical community [5]. With its spectrum of presentations, degrees of severity, wide variety of causes, and resurgence in incidence and public interest due to the SARS-CoV-2 pandemic, it is a challenging syndrome to diagnose and treat. However, there have been advances in the management of ARDS in recent years, and in combination with current therapies that have become mainstays of ARDS treatment, they have the potential to allow us to effectively address this common cause of morbidity and mortality for patients requiring critical care.

The diagnostic criteria of ARDS have evolved over the years, but currently rely on the 2012 Berlin definition, which was issued as an update to the American–European Consensus Conference (AECC) ARDS definition [6,7]. The Berlin definition [6] has four criteria:(1)Timing: “Within one week of a known clinical insult or new or worsening respiratory symptoms”(2)Chest Imaging (X-ray or Computed Tomography): “Bilateral opacities—not fully explained by effusions, lobar/lung collapse, or nodules”.(3)Origin of Edema: “Respiratory failure not fully explained by cardiac failure or fluid overload” with “objective assessment (e.g., echocardiography) to exclude hydrostatic edema if no risk factor present”.(4)Oxygenation:
Mild: 200 mm Hg < PaO_2_/FIO_2_ ≤ 300 mm Hg with PEEP or CPAP ≥ 5 cm H_2_OModerate: 100 mm Hg < PaO_2_/FIO_2_ ≤ 200 mm Hg with PEEP ≥ 5 cm H_2_OSevere: PaO_2_/FIO_2_ ≤ 100 mm Hg with PEEP ≥ 5 cm H_2_O

Despite advances in therapies, documented mortality rates range from 34–46% depending on the severity of the ARDS [8].

SARS-CoV-2, the cause of COVID-19, brought ARDS to the public’s attention, as roughly 6–10% of COVID-19 patients progressed to COVID-19-associated ARDS and required ICU care [9,10,11]. It was one of the many reasons that led to ICU beds running out in some areas during the initial wave of the pandemic and the subsequent variant waves [12,13]. COVID-19-associated ARDS and pneumonia meet the Berlin criteria for ARDS; as the literature suggests, they are very much their own diseases [14]. Furthermore, much like ARDS, they have multiple phenotypes that affect patient care and management [14]. Management of ARDS has also developed to meet the needs and ailments of patients, leading to new experimental therapies, with some showing greater efficacy than others. All treatments attempt to address or control a different aspect of the syndrome and pathogenesis. This, however, is a complex challenge.

## 2. Pathophysiology

The Berlin definition gives four criteria to diagnose ARDS, but it is best to understand ARDS as a multifaceted syndrome characterized by dysregulated systemic inflammation, increased endothelial and epithelial permeability, and diffuse alveolar damage, leading to alveolar edema, fibrosis, necrosis, and proteinosis [15,16,17]. This occurs over the course of three overlapping stages: exudative, proliferative, and fibrotic [18,19]. Not all patients proceed through each stage, and not all necessarily occur by themselves. There are a variety of risk factors, including alcohol abuse, obesity, patients who present with Acute Physiology and Chronic Health Evaluation scores > 16, increased use of transfusions, and those in septic shock [19].

The seven-to-ten-day exudative phase takes place after the initial insult, whether it was a pulmonary or extra pulmonary cause [20]. Inflammatory cells, such as macrophages and neutrophils, arrive and begin to migrate across the vascular and alveolar surfaces, disrupting the endothelial and epithelial barriers in the process. Disrupting these tight junctions increases the lung endothelial and alveolar epithelial permeability, allowing inflammatory cells, fluid, protein, and red blood cells to enter the airspace [21]. Further damage to type 1 and type 2 alveolar pneumocytes, loss of surfactant, and deposition of cellular debris along the alveoli airspace all impede the ability of the body to manage this alveolar edema. As a result, the alveolar airspace begins to fill with this protein and cell-rich fluid, leading to hyaline membrane formation, a histological hallmark of ARDS [15,16]. Other factors that play a role in this phase include lung epithelial glycocalyx shedding, vascular microthrombi, and platelets serving themselves as proinflammatory cytokines. Imaging at this stage may reveal the “bilateral opacities” indicative of alveolar edema, and the patient may require an ICU admission.

The proliferative phase takes place over the following days to weeks and is the body’s way of “recalibrating” itself to repair [18,20]. Anti-inflammatory cytokines facilitate neutrophil apoptosis and clearance, macrophages clear cellular debris via phagocytosis, type 2 pneumocytes begin to proliferate and differentiate into type 1 pneumocytes, the integrity of the epithelium is reestablished, and fluid begins to be cleared via active transport and diffusion [21,22].

The fibrotic stage of ARDS and what leads to it taking place in only some patients has yet to be understood. Furthermore, the role of fibroblasts in ARDS remains unclear. They are active in secreting epithelial growth factors as early as the exudative phase, and the growth factors aid in recovery during the proliferative stage. However, if the proliferative phase is prolonged or impaired, growth factors can be present in excess, leading to the deposition of fibrotic tissue, ultimately resulting in lung fibrosis [18,21]. There was an increase in post-ARDS fibrosis during the COVID-19 pandemic, even with lung-protective ventilation techniques, furthering the discussion around the spectrum of ARDS and even phenotyping [15].

Due to the prognostic significance, understanding the heterogeneity of ARDS has been the subject of research. This “phenotyping” and “sub-phenotyping” of ARDS was demonstrated in randomized controlled trials by the National Institutes of Health–National Heart, Lung, and Blood Institute (NIH–NHLBI) ARDS Network, where they showed that one phenotype had worse clinical outcomes associated with more severe inflammation, shock, and metabolic acidosis when compared to the other phenotype they had in their study [3,23]. However, this “hyper-inflammatory phenotype” (Table 1) may be one of many [24]. Better understanding of a variety of factors that could be behind these phenotypes is a part of the “precision medicine” approach to ARDS, with the fields of metabolomics and epigenetics playing crucial roles [25]. Due to the heterogeneous and diverse presentation of ARDS, metabolites or genetic markers could be used to create a “fingerprint” for the phenotypes, aiding in diagnosis and recognition, providing insight into ARDS disparities we see in patient populations, and even helping drive targeted patient care and management in the future [25,26]. We further discuss the metabolomics approach to ARDS later in this paper.

## 3. Current Management Strategies

As discussed earlier, ARDS is a heterogeneous process characterized by a mismatch in ventilation–perfusion and increased shunting, resulting in hypoxemia [27]. Currently, no effective pharmacotherapies are available to treat or prevent ARDS. Therefore, ARDS management focuses on treating the underlying lung injury and providing supportive care with mechanical ventilation to minimize Ventilator-Induced Lung Injury (VILI) and to manage refractory hypoxemia when present [27,28].

Lung protective ventilation, which refers to low tidal volume ventilation (LTVV) of 4–8 mL/Kg predicted body weight and plateau pleasure of <30 cm H_2_O, remains the keystone of managing ARDS. LTVV is thought to help minimize VILI by preventing tidal hyperinflation [29]. In a randomized clinical trial called the ARDSnet study, Brower et al. demonstrated that patients treated with lower tidal volumes (mean tidal volumes of 6.2 ± 0.8 mL per Kg of predicted body weight) had a significant reduction in mortality when compared to patients treated with traditionally high volumes (11.8 ± 0.8 mL per Kg of predicted body weight [30].

Another key element of lung protective ventilation is to provide positive end-expiratory pressure (PEEP) to improve hypoxemia and limit cyclic atelectasis [29]. The exact definition of what constitutes high vs. low PEEP has yet to be established. Most trials utilize the lower PEEP/higher FiO_2_ or higher PEEP/lower FiO_2_ values published by the ARDSnet group [30,31]. The work of Briel et al. showed a reduction in relative mortality with higher PEEP when compared with lower PEEP, and a similar adverse effects profile was observed in the two groups [31].

Given the simplicity of use and possible clinical benefits, non-invasive modes of ventilation have been proposed as first-line intervention for patients with mild to moderate ARDS; however, there remains a lack of data on clinical outcomes in severe hypoxemia, and the ideal interface for non-invasive ventilation remains unclear [32].

Modes of ventilation used in patients with ARDS can be classified as traditional and non-traditional. Traditional, more commonly used ventilation modes include pressure-controlled ventilation (PCV) or volume-controlled ventilation (VCV). In addition, a strategy called inverse ratio ventilation (IRV) may be applied to either PCV or VCV modes, where it essentially reverses the inspiratory to expiratory (I:E) ratio [33]. Alternative or non-traditional modes of ventilation include airway pressure release ventilation (APRV) and high-frequency oscillatory ventilation (HFOV). Different trials evaluated the use of such modes and found no evidence supporting the use of non-traditional ventilation modes over traditional ones [34,35].

Non-pharmacological interventions, such as prone positioning, have also been utilized in ARDS patients. Prone positioning is thought to improve oxygenation by reducing the ventral to dorsal transpulmonary pressure difference, ventilation–perfusion mismatch, and lung compression [36]. There have been more emerging data on the effect of prone positioning in the wake of the COVID-19 pandemic. A study in New York by Caputo et al. showed that non-intubated hypoxemic patients with suspected COVID-19 benefited from prone positioning, with a significant increase in SpO2 5 min after prone positioning [37]. Prone positioning for more than 12 h a day in patients with severe ARDS holds a strong recommendation in the American Thoracic Society/European Society of Intensive Care Medicine/Society of Critical Care Medicine clinical practice guidelines [38].

Another non-pharmacological intervention is the use of Extracorporeal Membrane Oxygenation (ECMO). ECMO is an extracorporeal life support modality that enables temporary support in pulmonary and/or cardiac failure that is resistant to standard therapy [39]. The British Thoracic Society guidelines recommend using ECMO in certain patients with severe ARDS who are on lung protective ventilation with a Murray Score >3 or pH < 7.2 due to uncompensated hypercapnia [40]. However, there are no firm recommendations for or against ECMO in severe ARDS in ATS/European Respiratory Society (ERS)/SCCM guidelines [38].

A variety of pharmacological agents have been studied in ARDS. Neuromuscular Blockers (NMBs) are thought to help in ARDS by decreasing the inflammatory burden of ARDS and by improving patient–ventilator synchronization by relaxing the smooth muscles. Patient–ventilator dys-synchrony is a common issue in mechanically ventilated patients and occurs when there is a mismatch between the ventilatory needs of the patients and the amount of ventilation delivered [41]. There has been conflicting evidence on the use of NMBs in ARDS. In 2010, the ACURASYS trial demonstrated the beneficial role of NMBs in ARDS by showing a lower 90-day mortality in the cisatracurium arm (30.8%) vs. the control arm (44.6%), and both arms had similar rates of ICU-acquired weakness [42]. A more recent trial called the Rose trial concluded that there was no significant difference in 90-day mortality between patients on early continuous cisatracurium infusion versus patients on a lighter sedation strategy [43].

Inhaled pulmonary vasodilators are used because they selectively dilate the arteries that perfuse well-ventilated lung units, resulting in greater oxygenation due to better ventilation/perfusion (V/Q) matching. Examples include inhaled nitric oxide (iNO) and inhaled prostacyclins (e.g., epoprostenol and iloprost). In patients with ARDS and refractory hypoxemia, pulmonary vasodilators have been shown to improve oxygenation; however, there are no beneficial effects on mortality, and these outcomes seem similar across the different vasodilators [44,45,46].

Due to their anti-inflammatory activity, multiple trials have examined the benefits of corticosteroids in managing ARDS. Recent data have shown improved outcomes, including mortality benefits in patients with moderate to severe ARDS, both in COVID-19 and non-COVID patients [47,48]. Currently, the Society of Critical Care Medicine (SCCM) and the European Society of Intensive Care Medicine (ESICM) provide a conditional recommendation for corticosteroid use in patients with ARDS and a PaO_2_/FiO_2_ ratio < 200. However, given the availability of newer data [47], an updated recommendation from these societies is expected in the future.

Given the rapid progression and high mortality associated with ARDS, antibiotics are frequently started empirically. However, due to prior exposure to antibiotics and the change in the lung microbiome, ARDS is often complicated by multidrug-resistant organisms, such as methicillin-resistant Staphylococcus aureus, *Pseudomonas aeruginosa*, and *Acinetobacter* [49], which are associated with worsened outcomes [50,51].

## 4. Pharmacotherapy in ARDS; Recent Trials

Due to the high morbidity and mortality associated with ARDS, a pharmacological cure has been sought for decades; however, this has largely been unsuccessful. Below, we review some of the important pharmacological trials conducted on patients with ARDS.

### 4.1. Beta Agonists

Previous studies have demonstrated that beta agonists can enhance the resolution of pulmonary edema and maintain the stability of the alveolar capillary membrane in baseline conditions [52,53,54,55], leading to the hypothesis that they may protect against lung injury and assist with alveolar fluid clearance in ARDS. Although some older studies have supported this theory and retrospective analyses of mechanically ventilated patients have indicated improved outcomes with beta agonists [56,57], subsequent larger randomized controlled trials (RCTs) have undermined their use in ARDS. The initial β-Agonist Lung Injury (BALTI) trial found improved lung function with systemic administration of beta-2-adrenoreceptor agonists in patients with ARDS but lacked statistical power to detect improvement in mortality [58]. However, the subsequent BALTI-2 trial, which evaluated patients administered with salbutamol by infusion, found no improvement in mortality and noted adverse events (particularly cardiovascular), indicating a net harm associated with the treatment [59].

In 2017, a randomized clinical trial [60] investigated the safety and feasibility of administering a combination of ICS and beta agonist early to prevent the onset of ARDS in patients at high risk of the condition. Fifty-nine patients with at least one known risk factor for developing ARDS, a Lung Injury Prediction Score ≥ 4, and acute hypoxemia were randomly assigned to receive either nebulized budesonide and formoterol or a placebo. The study’s primary endpoint was the change in the oxygen saturation divided by the fraction of inspired oxygen (S/F) ratio over five days as an indicator of improved oxygenation. The treatment group experienced a greater improvement in oxygenation, as measured by an increase in the S/F ratio. Another recent prospective study of 100 patients [61] reported similar findings, where inhaled salbutamol either as monotherapy or combined with ICS reduced the incidence of ARDS in hospitalized patients at risk of developing the condition.

In conclusion, limited evidence supports the hypothesis that inhaled LABAs can help prevent severe lung injury. Larger randomized trials focusing on early treatment with LABAs are needed. Systemic beta agonists are not tested in major trials given the risk of hypotension and systemic adverse events.

### 4.2. Sivelestat

Sivelestat sodium is a selective inhibitor of neutrophil elastase (NE) and has been studied in ARDS clinical trials for the past two decades with mixed results. NE plays an important role in the development and progression of ARDS. Previous research has indicated that NE can raise pulmonary vascular permeability, cause proteolysis of pulmonary tissue, and boost the production of leukocyte chemotactic factors, all of which can contribute to lung injury in a synergistic manner [62]. Studies conducted earlier have established a correlation between plasma NE levels and the extent of lung injury in animal models, as well as in patients with acute respiratory distress syndrome [63]. Consequently, blocking or inhibiting NE synthesis with antagonists could be a viable approach to treating ARDS. While widely used in Japan [64], the efficacy of sivelestat remains controversial. Phase III and IV trials conducted in Japan showed that sivelestat contributes to early weaning from the ventilator [65,66]. Furthermore, a large retrospective study performed using the Japanese nationwide administered database found that three-month mortality was significantly lower in the sivelestat group, and administration of sivelestat within seven days of admission may improve survival [67]. However, in international multicenter phase II studies, no significant difference in ventilator-free days was observed, and a higher 180-day mortality was noted in sivelestat-treated patients [68]. These discrepancies are difficult to explain; however, studies suggest that a serum procalcitonin level of 0.5 ng/mL or higher [69] and high red blood cell counts may indicate efficacy in specific ARDS populations [70].

In 2017, Pu et al. conducted a systematic search in multiple databases for RCTs that evaluated the effect of sivelestat sodium in patients with ARDS [71]. Six RCTs, comprising 804 patients with ARDS, were included. The results showed no significant difference in the risk of 28 to30-day mortality between sivelestat and the control. Additionally, sivelestat treatment had no significant impact on ventilation days, P/F ratios, or ICU stays. However, sensitivity analysis suggested that sivelestat therapy may impact the P/F level in ARDS patients. The meta-analysis found that while sivelestat therapy may increase the P/F level, it appears to have little or no effect on 28 to30-day mortality, ventilation days, and ICU stays.

There is a dearth of RCTs evaluating the use of sivelestat sodium in ARDS. More randomized studies are needed to confirm the findings of the recent retrospective studies and determine the optimal dosing and timing of sivelestat in ARDS.

### 4.3. Simvastatin

Simvastatin, a medication commonly used to lower cholesterol levels, works by inhibiting hydroxy-methylglutaryl coenzyme A reductase and has been proposed to have potential therapeutic benefits in ARDS due to its anti-inflammatory properties. Simvastatin has been shown to suppress inflammatory responses in both in vitro and in vivo murine models of lung inflammation. Furthermore, simvastatin has been observed to reduce endothelial permeability in vitro [72] and in animal models of lung injury following intratracheal lipopolysaccharide (LPS) instillation [73].

In 2008, a randomized clinical trial investigated whether simvastatin would affect the development of acute lung injury in healthy human volunteers exposed to inhaled LPS, a model of acute lung inflammation. Participants were randomized to receive either simvastatin or a placebo, followed by inhalation of LPS four days later [74]. Pretreatment with simvastatin was found to decrease LPS-induced bronchoalveolar lavage fluid neutrophilia, myeloperoxidase, tumor necrosis factor-α, matrix metalloproteinases, and C-reactive protein (CRP) in both pulmonary and systemic compartments, indicating that simvastatin has anti-inflammatory effects in humans experiencing acute lung inflammation. Subsequently, a multicenter, double-blind clinical trial (HARP-2) was conducted to compare simvastatin with a placebo in patients who had developed ARDS within the previous 48 h [75]. The results of the study indicated no substantial difference between the groups in terms of the average number of ventilator-free days, the number of days free of non-pulmonary organ failure, or mortality at 28 days. Similarly, a randomized trial of ARDS treated with simvastatin found no significant reduction in mortality at 12 months [76]. In 2018, Calfee et al. conducted a secondary analysis using data from 539 patients enrolled in the HARP-2 trial [77], and clinical outcomes were compared across sub-phenotypes (hypo-inflammatory and hyper-inflammatory) and treatment groups. Although the original trial had found no difference in 28-day survival between the simvastatin and placebo groups, the secondary analysis revealed that survival rates varied significantly among patients stratified by treatment and sub-phenotype. Specifically, in the hyper-inflammatory sub-phenotype marked by higher levels of interleukin-6 and soluble tumor necrosis factor receptor-1, patients who received simvastatin had substantially higher 28-day survival rates than those who received a placebo, and a comparable pattern was noticed for 90-day survival.

Another secondary analysis of the HARP-2 study examined 28-day mortality and response to simvastatin according to baseline plasma IL-18 [78]. The researchers also examined the effect of simvastatin and rosuvastatin on monocyte-derived macrophages taken from healthy volunteers. The macrophages were pre-treated with either simvastatin or rosuvastatin before being stimulated with adenosine triphosphate (ATP) and LPS, and their secretions of IL-18 and IL-1β were measured. The results showed that patients with high baseline plasma IL-18 levels (≥800 pg/mL) had an increased risk of 28-day mortality. However, those patients who were given simvastatin had a lower probability of 28-day mortality compared to those given a placebo. Interestingly, simvastatin but not rosuvastatin reduced the secretion of IL-18 and IL-1β in stimulated macrophages. These findings indicate that baseline plasma IL-18 levels could be used to personalize treatment for ARDS patients by identifying those who may benefit from simvastatin therapy.

### 4.4. Vitamin D

Vitamin D deficiency is linked to a higher likelihood of admission to the intensive care unit and mortality in patients with pneumonia [79]. Vitamin D deficiency is prevalent among critically ill patients and is linked to negative outcomes [80]. Studies have also demonstrated Vitamin D to have anti-inflammatory effects [81], modulate cytokine responses [82], and promote the growth and differentiation of epithelial cells [83].

Earlier studies have established a correlation between Vitamin D levels and the duration of mechanical ventilation, functional status upon ICU discharge, outcomes after post-acute care hospitalization, and mortality rates in critically ill surgical patients [84,85,86,87]. Furthermore, high-dose Vitamin D supplementation improved clinical results in critically ill patients with low initial levels of 25-hydroxyvitamin D (25OHD) [88].

In 2022, a secondary analysis of the ALVEOLI study was carried out to explore the connection between vitamin D status and two outcomes: ventilator-free days and 90-day survival [89]. The study revealed that patients with vitamin D deficiency had a longer duration of mechanical ventilation and a greater likelihood of 90-day mortality. Patients with a 25OHD level below 20 ng/mL were predicted to undergo ventilation for three additional days compared to those with levels equal to or above 20 ng/mL. Patients with a 25OHD level below 10 ng/mL were likely to be ventilated for nine extra days and had a 34% higher risk of 90-day mortality than those with levels greater than 10 ng/mL.

In view of these data, randomized, controlled trials are needed to determine the clinical effects of vitamin D supplementation in ARDS patients.

## 5. Mesenchymal Stromal Cells

Mesenchymal stromal cells (MSCs) are spindle-shaped cells with multipotent differentiation capacity which can be extracted from various sources, including bone marrow, the umbilical cord, adipose tissue, etc. [90]. Recently, MSCs have been the center of attention after studies showed that they have anti-inflammatory properties with the potential to affect both the innate and adaptive immune systems [91,92]. Clinically, MSCs have been used for various diseases, including skin pathologies [93], systemic lupus erythematosus [94], HIV infection [95], and osteoarthritis [96], with promising results.

Given the immunomodulatory effects, MSCs are being extensively studied in immune-regulated respiratory pathologies. In a preclinical study on a bleomycin-induced fibrosis model, MSCs were shown to have cytoprotective properties with prevention of apoptosis and promotion of epithelial cell proliferation [97]. Preclinical data also suggests therapeutic benefits in asthma [98] and COPD [99].

The pathogenesis of ARDS is characterized by neutrophil-dependent lung injury. Neutrophil accumulation results in endothelial injury coupled with alveolar epithelial damage leading to alveolar edema. [21] MSCs have been shown to upregulate tumor necrosis factor stimulated gene 6 (TSG-6), which inhibits neutrophil migration via interaction with the chemokine CXCL8 [100]. MSCs also exert anti-apoptotic effects by inhibiting the Wnt/B-catenin pathway (Li), as well as suppressing the tumor necrosis factor-α (TNF-α) driven Fas/FasL pathway [101].

Multiple preclinical studies have been published in the last 15 years, reporting the safety and efficacy profile of MSCs in animal models [102,103,104]. Acute lung injury (ALI) in these models can be stimulated by infectious, chemical, or ischemic insults. A systematic meta-analysis by McIntyre et al. [105] on preclinical studies of MSCs in animal models showed that MSCs decrease the overall odds of mortality in animal models with ALI compared to control models without MSCs treatment. Importantly, this survival benefit effect was maintained regardless of type of the animal model, type of insult, source of MSCs, and the route of MSCs administration.

Based on preclinical data, MSCs were studied in phase 1 trials for assessment of safety. In the START trial [106], nine patients with ARDS were subjected to varying doses of MSCs (1 million cells/kg predicted body weight to 10 million cells/kg predicted body weight). No treatment-related adverse events were observed in any of the patients. In the dose escalation (100–400 million cells), phase 1 REALIST-trial [107], moderate to severe ARDS patients were treated with umbilical cord-derived MSCs. No dose-limiting toxicity was observed. The MUST-ARDS trial [108] was a phase 1/2 trial conducted in centers in the USA and the UK. The safety and efficacy of MSCs was studied using three different cohorts. To assess acute safety, cohort 1 was treated using 300 million cells and cohort 2 was treated with 900 million cells. Cohort 3 was randomized, double-blinded, and placebo-controlled, with a treatment group (infused 900 million cells) and a control group (placebo). Again, no treatment-related adverse events were observed in any of the cohorts. In cohort 3, use of MSCs was associated with improved 28-day mortality (25% vs. 45%) and ventilation-free days.

## 6. Gut–Lung Interaction in ARDS

The airway space is inhabited by non-pathogenic bacteria that originate from oropharyngeal sources and maintain lung immune hemostasis [109]. However, experimental animal models and human studies have shown enrichment of the pulmonary microbiome with gut-associated bacteria in ARDS [110,111]. Interestingly, this microbiome “dysbiosis” has been proposed to occur secondary to translocation from the gut rather than aspiration [111]. The alteration in microbiome is associated with increased interleukin 6 (IL-6) levels and worsened outcomes [112]. This gut–lung cross talk may represent another therapeutic target in the management of ARDS in the future, with fecal microbiota transplantation as a potential intervention [113].

## 7. Metabolomics and Future Directions

Metabolomics is the study of various low molecular weight (less than 1 kDa) metabolites in an organism at a point in time [114]. The metabolome is a downstream product of transcriptome and proteome [Figure 1] and has been regarded as the reflection of phenotype [114]. The goal of the qualitative and quantitative analysis of these metabolites is not only to understand the pathophysiology of the disease process, but also to predict therapeutic responses to interventions [115]. The use of metabolomics profiling has been well described in phenotyping Parkinson’s disease, diabetes, and cardiovascular pathologies [116,117,118].

As discussed earlier, ARDS is a heterogeneous entity with different phenotypes. Despite decades of research, it remains unclear why some patients progress to severe disease while others do not. Interestingly, despite the objective nature of Berlin’s criteria, there remains up to 30% inter-observer variability in the interpretation of chest X-rays consistent with ARDS [119]. Moreover, diffuse alveolar damage is only seen in 50% of autopsy tissue samples from patients clinically diagnosed with ARDS [120]. As such, metabolomics profiling is an attractive tool to phenotype and stratify patients with ARDS.

A variety of molecules are measured in metabolomics, including peptides, sugars, fatty acids, and steroids, and can be analyzed using nuclear magnetic resonance spectroscopy (NMR), gas chromatography mass spectrometry (GC-MS), and liquid chromatography mass spectrometry (LC-MS) platforms [121].

One of the first studies of metabolomics profiling in ARDS was performed by Schubert et al. [122] in 1998, in which they used GC-MS to compare nine metabolites in the exhaled breath of ventilated ARDS patients against ventilated surgical patients. They found that patients with ARDS had low levels of isoprene, a by-product of cholesterol metabolism hypothesized to have a thermo-protective effect against reactive oxygen species [123]. However, the study was limited by the small sample size and lack of validation. Applying a similar theme of using GC-MS on the exhaled breath of patients with ARDS versus non-ARDS ventilated patients, Bos et al. [124] identified 3-methylheptane, octane, and acetaldehyde as biomarkers for ARDS. However, on a subsequent validation study, the biomarkers only had moderate diagnostic accuracy [114].

Studies have also been conducted on bronchoalveolar lavage fluid (BALF) samples using H-NMR [125] and LC-MS [126], identifying increased lactate [125,126], branched- chain amino-acids and decreased ethanol [125], and phosphatidylcholine levels [126] as a metabolomics fingerprint associated with ARDS. However, these studies have been limited by the small sample size and lack of validation. Some of these studies have been criticized for using healthy non-ventilated patients as controls, since it has been shown that mechanical ventilation impacts metabolites [127].

Moving forward, it is important that clinical trials account for the different phenotypes of ARDS, as biological heterogeneity in ARDS has significant clinical implications. Initial data on mesenchymal stem cells is promising, and by integrating metabolomics profiling in clinical studies, the goal of “personalized medicine“ may not be unrealistic.

## Figures and Tables

**Figure 1 diagnostics-13-01528-f001:**
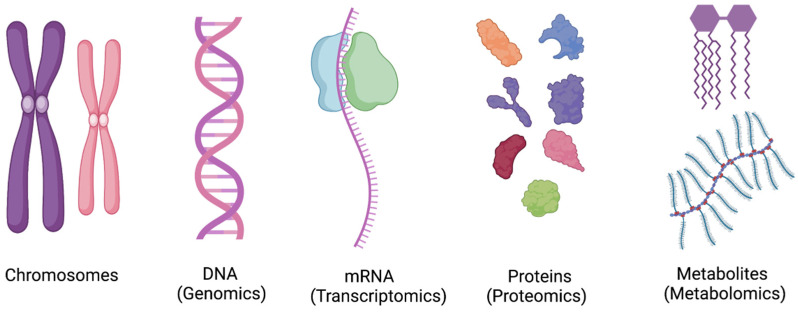
The concept of “Omics” in scientific research. Metabolomics is the newest addition, focusing on analyzing low molecular weight compounds. DNA = Deoxyribonucleic acid, RNA = Ribonucleic acid.

**Table 1 diagnostics-13-01528-t001:** Differences between hyperinflammatory and hypo-inflammatory phenotypes of ARDS.

Hyperinflammatory Phenotype Compared to Hypo-Inflammatory Phenotype
High levels of IL-6, IL-8, sTNFr-1
Lower serum bicarbonate levels
Low protein C levels
More profound hypotension
Higher mortality
Less ventilator-free days
Lower mortality with higher PEEP
Improved mortality with fluid conservative strategy

IL-6 = Interleukin 6, IL-8 = Interleukin 8, sTNFr-1 = Soluble Tumor Necrosis Factor 1 receptor.

## Data Availability

Not applicable.

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
