# Peer review of "Acute Respiratory Distress Syndrome; A Review of Recent Updates and a Glance into the Future"

_diagnostics, 2023, doi:10.3390/diagnostics13091528_

Round 1

Reviewer 1 Report

Review report
      The review article entitled “Acute Respiratory Distress Syndrome; A review of recent updates and a glance into the future” has been prepared in a scientific manner and easy to understand. However, some minor edits could improve the quality and coverage of the article.

1-    Since some of the affiliation addresses of authors are same, it could be unified. Also the title of authors would not be posted with the name.
2-    Metabolomics, immunological, herbal medicine and epigenetic aspects were not sufficiently covered through the manuscript in the pathogenesis or in pharmacotherapy sections. Suggest adding some more details from the recent articles.
3-    Add section to the manuscript to address the role of gut-lung-axis microbiome and its metabolites on progress of ARDS and how could be therapeutic.
4-    How the misuse of antibiotics and other medications affect the progress of ARDS was not mentioned in the article.

Author Response

We would like to thank the reviewer for the valuable feed back.

  1. We have removed the title of authors that was previously associated with the names.
  2. We added more insight in the pathogenesis part of article on metabolomics and epigenetic aspects
  3. A section on gut-lung axis was added and recommended by the reviewer
  4. The effect of antibiotics on ARDS was included in the manuscript

Reviewer 2 Report

Acute Respiratory Distress Syndrome; a review of recent up-dates and glance into the future

This is an interesting and well written review on acute respiratory distress syndrome. The authors describe exhaustively the main characteristics of this syndrome, the novelties in treatment and some interesting information on future research prospects. Furthermore, the authors highlights the importance of the recent COVID-19 pandemic on the recent re-ignited interest on this syndrome and its treatment.  In fact, precisely the difficulty in managing patients in severe respiratory failure during the pandemic, has reopened a debate on all aspects of this syndrome, starting with the definition itself and which deserves to be explored.

As correctly described by the authors in the introduction, a certain number of COVID-19 patients developed severe ARDS syndrome which leaded to extreme treatments such as ECMO; at the same a significant number of patients presented all the Berlin’s criteria (i.e. bilateral infiltrates not of hydrostatic origin with associated severe and refractory shunt) but they did not present the impairment of respiratory mechanics that is one of most difficult to treat features of ARDS (Gattinoni L, Coppola S, Cressoni M, Busana M, Rossi S, Chiumello D. Covid-19 does not lead to a “typical” acute respiratory distress syndrome. Am J Respir Crit Care Med. 2020;201:1299–1300).

Given the great variety of pathophysiological pictures with which ARDS occurs within the same disease, a debate has started on the opportunity to revise the definition of the syndrome (Ranieri et al. Rethinking ARDS after COVID-19. If a better definition is the answer, what is the question? Am J Rep Crit Care Med 2022). Some authors were very critical on the need for a revision of the definition (Tobin MJ, Defining ARDS (again): a plea for honesty. Am J Crit Care Med 2022). In a recent prospective article, Martin J Tobin, observed how much the Berlin Task Force Definition could be dangerous in the context of a respiratory pandemic, encouraging early endotracheal intubation in patients meeting criteria for ARDS  (Tobin MJ, ARDS: hidden perils of an overburdened diagnosis, Crit Care 2022). 

Despite the differences, after the experience of the pandemic, most authors agree that ARDS is a syndrome that requires treatment adapted to the pathophysiology of the individual patient and promotes a more flexible approach that should focus treatments on the characteristics of the patient (e.g. ventilatory inequality, inflammatory response, mechanical alteration…). All this debate, which justifies various innovative and interesting therapeutic approaches, brought to light by the pandemic, is little explored in the review and deserves a mention. 

For example, the pandemic has highlighted how, in some cases, the non-invasive ventilatory approach could be safer and more profitable for the patient rather than early intubation, reducing the risk of evolution of lung damage induced by the patient himself (Patient Self-inflicted Lung Injury, P-SILI) (Grieco DL et al. Non-Invasive ventilatory support and high-flow nasal oxygen as first-line treatment of acute hypoxemic respiratory failure and ARDS, Intensive Care Med 2021).

Minor comments: 

Page 4 – Paragraph 2 “Non-pharmacologica interntions, such as prone positioning…”

The prone position certainly improves ventilation/perfusion of the lung due to the different distribution of ventilation and pulmonary perfusion, and this is particularly true in covid patients in whom severe pulmonary circulation vasoplegia has been demonstrated. Nevertheless, some studies have shown different concentration of NO-Synthetases form dorsal and ventral lung portion that could justify the positive effect of prone position independently to the etiology of ARDS (Oxygen and Life on EarthAn Anesthesiologist’s Views on Oxygen Evolution, Discovery, Sensing, and Utilization Anesthesiology. 2008;109(1):7-13. doi:10.1097/ALN.0b013e31817b5a7e).

Page 5 – Paragraph 4.1

It should be highlighted that almost all the trials cited utilized systemic b-Agonist which is a route of administration not without risks and with many potentially harmful cardiovascular effects. The administration of these drugs by inhalation could be a promising strategy, both for the effects on membrane stability and for the possible protection from P-SILI thanks to the bronchodilator effect.

Author Response

We thank the reviewer for their extensive and valuable review.

We have included more details on the phenotypes of ARDS and need for individualized care of these patients with ARDS

We have included reference for NIV in ARDS

We mention that the trials on Beta agonist were not systemic (due to risk of adverse events) as mentioned by reviewer.

Reviewer 3 Report

Based on the Rivers trial of early goal-directed therapy (N Engl J Med. 2001 Nov 8;345(19):1368-77), central venous oxygen saturation (cvO2%) emerged as a resuscitation target.  This was recommended for over a decade until the PROCESS, PROMISE, and ARISE trials demonstrated that cvO2% monitoring was unnecessary.  However,  this concept has a certain physiologic appeal and has been advocated by some authors to sort out which type of shock a patient has.  (Gattinoni 2013 https://pubmed.ncbi.nlm.nih.gov/23514343/) and reemerged with the Covid pandemic.

The author should discuss in brief how ARDS Berlin definition meets COVID pneumonia clinical presentation and if different clinical phenotypes of Covid pneumonia, as described by Gattinoni (DOI: 10.1186/s13054-020-02880-z), need different biological measurements and targets, as cvO2% or etCO2/PaCO2to titrate protective ventilation

Author Response

We thank the reviewer for their valuable feedback and agree with them regarding their input.

We have added the recommended reference and a brief introduction into the different phenotypes of ARDS requiring different physiological targets for management